# V-Domain Ig Suppressor of T Cell Activation (VISTA) Expression Is an Independent Prognostic Factor in Multiple Myeloma

**DOI:** 10.3390/cancers13092219

**Published:** 2021-05-06

**Authors:** Pim Mutsaers, Hayri E. Balcioglu, Rowan Kuiper, Dora Hammerl, Rebecca Wijers, Mark van Duin, Bronno van der Holt, Annemiek Broijl, Walter Gregory, Sonja Zweegman, Pieter Sonneveld, Reno Debets

**Affiliations:** 1Department of Hematology, Erasmus MC Cancer Institute, 3015 GD Rotterdam, The Netherlands; m.vanduin@erasmusmc.nl (M.v.D.); b.vanderholt@erasmusmc.nl (B.v.d.H.); a.broyl@erasmusmc.nl (A.B.); p.sonneveld@erasmusmc.nl (P.S.); 2Department of Medical Oncology, Erasmus MC Cancer Institute, 3015 GD Rotterdam, The Netherlands; h.balcioglu@erasmusmc.nl (H.E.B.); d.hammerl@erasmusmc.nl (D.H.); r.wijers@erasmusmc.nl (R.W.); j.debets@erasmusmc.nl (R.D.); 3SkylineDx BV, 3062 ME Rotterdam, The Netherlands; R.Kuiper@skylinedx.com; 4Clinical Trials Research Unit, Leeds Institute of Clinical Trials Research, Leeds LS2 9JT, UK; w.gregory@leeds.ac.uk; 5Department of Hematology, Amsterdam University Medical Center, VUMC, 1081 HV Amsterdam, The Netherlands; s.zweegman@amsterdamumc.nl

**Keywords:** multiple myeloma, tumor immunology, immune therapy, V-domain Ig suppressor of T cell activation (VISTA), immune checkpoints

## Abstract

**Simple Summary:**

Multiple myeloma (MM) is characterized by loss of anti-tumor T-cell immunity. The precise mechanisms by which malignant plasma cells escape T-cell immunity are unknown, although upregulation of checkpoint molecules is seen in progressive disease. The aim of our study was to investigate mechanisms of escape from T-cell immunity. We observed that the expression of V-domain Ig suppressor of T cell activation (VISTA) in the tumor microenvironment is an independent prognostic factor for survival in MM and its major cellular source is tumor infiltrating CD11B+ cells. The combination of high VISTA expression in the tumor combined with low infiltration of CD8+ cells compared to the surrounding stromal tissue is significantly associated with poor survival. These finding have identified VISTA as an interesting target for inhibition to circumvent escape of T-cell immunity.

**Abstract:**

Multiple myeloma (MM) is characterized by loss of anti-tumor T cell immunity. Despite moderate success of treatment with anti-PD1 antibodies, effective treatment is still challenged by poor T cell-mediated control of MM. To better enable identification of shortcomings in T-cell immunity that relate to overall survival (OS), we interrogated transcriptomic data of bone marrow samples from eight clinical trials (*n* = 1654) and one trial-independent patient cohort (*n* = 718) for multivariate analysis. Gene expression of V-domain Ig suppressor of T cell activation (VISTA) was observed to correlate to OS [hazard ratio (HR): 0.72; 95% CI: 0.61–0.83; *p* = 0.005]. Upon imaging the immune contexture of MM bone marrow tissues (*n* = 22) via multiplex in situ stainings, we demonstrated that VISTA was expressed predominantly by CD11b+ myeloid cells. The combination of abundance of VISTA+, CD11b+ cells in the tumor but not stromal tissue together with low presence of CD8+ T cells in the same tissue compartment, termed a high *VISTA-associated T cell exclusion* score, was significantly associated with short OS [HR: 16.6; 95% CI: 4.54–62.50; *p* < 0.0001]. Taken together, the prognostic value of a combined score of VISTA+, CD11b+ and CD8+ cells in the tumor compartment could potentially be utilized to guide stratification of MM patients for immune therapies.

## 1. Introduction

Disease progression in multiple myeloma (MM) is associated with loss of immune-mediated control of plasma cell growth. In MM, such immune evasion is characterized by compromised anti-tumor CD8+ T-cell responses [1,2]. Mechanisms that underly hampered CD8+ T-cell responses are not yet defined in MM, but have been thoroughly described in other cancers such as melanoma, non-small cell lung cancer, and breast cancer. T-cell evasive mechanisms that occur in the latter types of cancers include limited influx and migration (i.e., lack or down-regulated expression of chemo-attractants and/or adhesion molecules), antigen recognition (i.e., lack or down-regulated expression of molecules involved in antigen processing and/or presentation), and/or function of CD8 T-cells (i.e., presence of immune-suppressor cells, altered expression of immune or metabolic checkpoints, and/or activation of oncogenic pathways) [3,4]. Along these lines, diminished anti-tumor CD8+ T-cell responses in MM have indeed been linked to the presence of immune-suppressive cells such as regulatory T-cells [5] and myeloid-derived suppressor cells [6,7] as well as the expression of immune checkpoints by tumor- and stromal cells [8,9].

Increased expression of the immune checkpoint molecules PD1 and PDL1 by tumor-infiltrating T-cells and plasma cells, respectively, has been observed in MM patients who relapsed after treatment when compared to patients in longer remission; and also in minimal residual disease (MRD)-positive patients compared to MRD-negative patients [8]. Nevertheless, trials investigating PD1 blockade with nivolumab in MM have been disappointing so far, with monotherapy resulting in stable disease as the best response in all (*n* = 27) but one patient [10]. Notably, combination therapies with either nivolumab or pembrolizumab and an immunomodulatory drug resulted in objective response rates of 44 and 60%, respectively, including complete remissions [11,12,13], however, these responses occurred at the cost of excess toxicity and increased treatment-related mortality. The use of these combinations has therefore largely been abandoned. In extension to PD1, other immune checkpoint inhibitors have also been studied in MM such as TIM-3, LAG3, OX40, and GITR blockade. These treatments restore autologous anti-myeloma T-cell responses in vitro [9], and are currently under development for patient testing.

In the present study, we assessed the contribution of the immune micro-environment to disease progression in MM via analysis of expression as well as tissue localization of genes related to T-cell evasion [4,14]. To this end, we performed expression analysis of 366 immune-related genes that cover the reported mechanisms of T-cell evasion in bone marrow aspirates of 2372 patients, in situ stainings of candidate gene products in 22 patients, and correlative analyses between gene as well as contextual outcomes and overall survival (OS) of MM patients.

## 2. Results

### 2.1. Transcriptomics of Bone Marrow Samples Identifies VISTA as an Independent Prognostic Factor for MM

To identify immune-related genes that have prognostic value in MM, we analyzed transcriptomics data from pre-treatment bone marrow samples, purified for plasma cells, from large patient cohorts (workflow of gene expression analysis is schematically presented in Figure 1).

We started with seven lists of T-cell evasion-related genes (*n* = 366 genes, as described by Hammerl et al. [15]), and performed a 3-staged gene expression analysis. First, ridge regression COX models for OS yielded five lists of genes (i.e., cell death, immune checkpoints, metabolic checkpoints, oncogenic signaling pathways, tumor micro-environment, *n* = 328 genes) that were statistically significant when applied to both the discovery as well as validation sets (*n* = 1045 and 609 patients, respectively, see Table 1 and Materials and Methods for details). Second, individual gene analysis yielded 72 genes that showed a significant association with OS in the discovery set and a remaining six genes in the validation set. These six genes were: microtubule associated protein 1 light chain 3 alpha (MAP1LC3A) (HR: 0.80; 95% CI: 0.71–0.90; *p* = 0.01); DNA damage regulated autophagy modulator 1 (DRAM1) (HR: 0.79; 95% CI: 0.70–0.90; *p* = 0.017); V-domain Ig suppressor of T-cell activation (VISTA) (HR: 0.76; 95% CI: 0.67–0.81; *p* = 0.001); mitochondrial outer membrane import complex protein 2 (MTX2) (HR: 1.24; 95% CI: 1.07–1.44; *p* = 0.003); gamma-interferon-inducible protein 16 (IFI16) (HR: 1.37; 95% CI: 1.21–1.58; *p* = 0.0005); and platelet endothelial cell adhesion molecule (PECAM1) (HR: 0.73; 95% CI: 0.64–0.82; *p* < 0.0001). Third, multivariate validation in the CoMMpass cohort of these six genes, correcting for staging and autologous transplantation, yielded VISTA as a single gene with a significant HR of 0.75 (95% CI: 0.62–0.92; *p* = 0.005). Detailed data of the above three steps are listed in Table 2, Table 3 and Table 4, and the survival advantage of high VISTA gene expression is illustrated in forest plots in Figure 2 using the cohorts of the discovery as well as validation sets.

### 2.2. MM Bone Marrow CD11b+ Cells, but Not Plasma Cells, CD4, or CD8 T-Cell Subsets nor CD163+ Cells, Express VISTA Protein

In order to extend the gene expression data and localize VISTA+ cells, we looked into VISTA protein expression in bone marrow samples from MM patients (workflow of immune stainings is schematically presented in Figure 1). These were samples taken from patients that were treated in the HOVON-87/NMSG18 trial, a randomized, phase-III trial investigating melphalan, prednisolone, and thalidomide compared to melphalan, prednisolone, and lenalidomide in newly diagnosed, elderly MM patients (Appendix A). Our initial immune fluorescence stainings clearly showed distinctive VISTA+ cells in the micro-environment of MM, however, there was no concurrence between the density of VISTA+ cells and levels of VISTA gene expression in our exploratory cohort of 22 patients (Appendix A). Along this observation, we found that CD138+ plasma cells were negative for VISTA protein in all analyzed MM samples (Figure 3A). To address the apparent discrepancy between analyses of gene expressions and immune stainings, we tested whether the purity of FACSorted bone marrow samples for CD138+ cells presented a confounding factor. To this end, we used data from the HOVON65/GMMG-HD4 and HOVON87/NMSG18 cohorts for which percentages of CD138+ cell purity were available (median in both cohorts: 92; and 95% CI: 78–100 and 77–99, respectively), and demonstrated that these percentages neither affected OS directly nor the above-mentioned association between VISTA and OS (Appendix A). Moreover, we assessed the relative distribution of 21 immune cell populations in the bone marrow samples that were used for gene expression analysis and again revealed significant presence of CD138-negative cells (Appendix A). Collectively, the above outcomes point toward the direction of immune cells, other than plasma cells, present in the micro-environment of MM as being a prominent source for VISTA expression.

In a next series of experiments, we set out to identify which cell type(s) expressed VISTA using multiplex immunofluorescence against CD4, CD8, CD11b, CD163, FOXP3, and VISTA. With this panel, we demonstrated that CD11b+ cells, but not other immune cells, represent the predominant cellular source of VISTA (Figure 3B–F). Images with three colors (VISTA in green; immune marker in red; and DAPI in blue) clearly show VISTA-expressing cells only in the case VISTA is combined with CD11b stainings (arrows in Figure 3F). The presence of VISTA-expressing CD11b cells as well as VISTA-negative T-cells (whether it be CD4, CD8, FOXP3) and VISTA-negative CD163 myeloid cells was confirmed in multiplex images (Figure 3G,H). It is noteworthy that besides CD11b+ cells, other non-immune cells in the biopsy, morphologically identified as erythrocytes and megakaryocytes, were also positive for VISTA (data not shown). For downstream analysis, we set up a second multiplex immune fluorescence panel incorporating the markers CD138, CD8, CD11b, and VISTA (Figure 3H).

### 2.3. Densities of VISTA+, CD11b+ Cells or Distances between VISTA+, CD11b+, and CD8+ T-Cells in Different Tissue Compartments Do Not Correlate to OS in MM

To test the prognostic value of VISTA+, CD11b+ cells, we assessed the densities of VISTA+ and CD11b+ cells in both tumor as well as stromal compartments of bone marrow tissues, and tested whether cellular densities in either compartment were different between MM patients with low versus high OS (*n* = 22, median cut-off, Figure 4). Densities of VISTA, CD11b, CD8, or CD138 single positive cells were taken along as controls. Our results showed that densities of VISTA+, CD11b+ cells, or those of control cell types were non-different between the two MM patient groups irrespective of tissue compartment. In addition, we investigated distances between VISTA+, CD11b+ cells, and CD8 T-cells in both tissue compartments from MM patients with low versus high OS (Figure 5). Again, analyses revealed that neither VISTA+, CD11b+ cells, nor control cell types were positioned at different distances relative to CD8 T-cells when comparing MM patients with low versus high OS.

### 2.4. High Density of VISTA+, CD11b+ Cells in Tumor over Stroma Combined with Low Density of CD8+ T-Cells in Tumor over Stroma Associates with Short OS in MM

Given the inability of single tissue contextual parameters to distinguish MM patients according to OS, we looked in more detail into inter-relationships between different contextual parameters. We observed that preferential localization of CD8+ T-cells in the tumor but not stromal compartment (i.e., high ratio between tumoral and stromal CD8 T-cell densities) significantly correlated with preferential localization of VISTA+ and CD11b+ cells in the same compartment (i.e., high ratio between tumoral and stromal VISTA+, CD11b+ cell densities) (Figure 6A). Strikingly, patients with low and high OS harbored low and high preferential tumoral CD8 T-cell densities, respectively, whereas corresponding tumoral VISTA+ and CD11b+ cell densities were non-different (Figure 6A: low (black dots) and high OS (red dots) positioned below and above the regression line, respectively). Microscopic images in Figure 6B illustrate that patients with low OS harbored most CD8+ T-cells in the stromal compartment and most VISTA+ and CD11b+ cells in the tumor compartment. In contrast, patients with high OS harbored most CD8+ T-cells in the tumor compartment and most VISTA+ and CD11b+ cells in the stromal compartment.

When translating the above findings into Kaplan–Meier curves, we demonstrated that preferential localization of CD8 T-cells in the tumor compartment had a beneficial effect toward OS, whereas preferential localization of VISTA+ and CD11b+ cells in this compartment had an adverse effect toward OS (Figure 6C,D). Importantly, in case each parameter was used separately, these effects did not reach significance. The combination of these two parameters (i.e., preferential localization of VISTA+, CD11b cells in tumor multiplied by preferential localization of CD8 T-cells in stroma,), however, the parameter we termed *VISTA-associated T-cell exclusion* clearly divided MM patients with low versus high OS (Figure 6E, [HR: 16.60; 95% CI: 4.54–62.50; *p* < 0.0001]).

## 3. Discussion

In the current study, using large numbers of patients and employing multiple layers of validation, we have demonstrated that VISTA gene expression in bone marrow samples is an independent prognostic factor in patients with both newly diagnosed (NDMM) as well as relapsed refractory MM (RRMM). This is a novel finding and has not been reported before. In addition, we identified intra-tumoral CD11b+ cells, but not CD138+ plasma cells or other immune cells as the major cellular source of VISTA expression. Finally, we introduced the *VISTA-associated T-cell exclusion score,* which captures two tissue contextual parameters that relate to the tumoral abundances of VISTA+, CD11b+ cells as well as CD8+ T-cells, and which strongly associates with poor OS in MM.

VISTA is a checkpoint molecule, part of the Ig superfamily and is a recognized homologue to PDL1. Like other members of the B7-CD28 family, T-cells and antigen presenting cells express VISTA. Recent studies have identified potential VISTA ligands that are expressed by tumor and other stromal cells such as V-Set and Immunoglobulin Domain-Containing Protein-3 (VSIG-3) [16] and P-Selectin Glycoprotein Ligand 1 (PSGL1) [17]. Importantly, VISTA-Fc fusion protein and overexpression of VISTA suppress T-cell recruitment [18], proliferation, and cytokine production [19,20]. Indeed, in vivo mouse and human VISTA blockade were found to enhance anti-tumor T-cell responses [19,21]. Interestingly, in the context of high PDL1 expression, blocking VISTA resulted in enhanced T-cell response, even in the absence of high VISTA expression [21]. It is noteworthy that in our gene expression analyses, PDL1 reached a statistical trend, suggesting the potential existence of a biological interrelationship between the two immune checkpoints. An explanation for our finding that gene expression of the T-cell inhibiting molecule VISTA positively associates with OS in MM may be that many T-cell evasive mechanisms, amongst which the expression of immune checkpoints often take place as part of a negative feedback loop following initial T-cell activation [3]. Along this line, the presence of immune checkpoint molecules can be associated with improved outcome [22], and can be related to higher frequencies of intra-tumoral CD8+ T-cells [23].

Once we established VISTA gene expression as an independent prognostic factor for OS in NDMM and RRMM, we determined the cellular source of VISTA. This analysis revealed significant presence of CD138-negative cells in bone marrow aspirates, however, the purity for CD138 cells did neither affect OS nor the association between VISTA and OS according to the data of two patient cohorts (Appendix A). We did not have enrichment details for the other six cohorts, however, methods for FACSorting as well as HRs for VISTA of all individual trials were similar, strongly suggesting that the observed prognostic value of VISTA gene expression is highly linked to non-CD138 cells present in the immune micro-environment of MM. By extension, immunofluorescence images demonstrated high levels of VISTA protein expression not on CD138 plasma cells, but instead on CD11b+ myeloid cells. It is noteworthy that VISTA protein expression was not detected on intra-tumoral CD4 or CD8 T-cells, FOXP3+ CD4 T-cells, nor CD163+ macrophages. Moreover, we did not observe VISTA expression on T-cells from five healthy donor-derived bone marrow samples (data not shown). In line with our immunofluorescence data, VISTA expression on CD11b+ cells has been previously observed in mice, where VISTA expression was upregulated in CD11b+ myeloid cells in the tumor when compared to peripheral blood [21]. In addition, in a mouse model of colorectal cancer, it was observed that CD11b+ myeloid-derived suppressor cells were the predominant cellular source of VISTA [24].

Several studies have reported on VISTA expression and its association with survival in certain non-MM cancers. For instance, one study observed high VISTA protein expression to be associated with high abundance of CD8+ tumor-infiltrating lymphocytes and favorable outcome in hepatocellular carcinoma [25], while another study observed high VISTA protein expression to be clinically unfavorable in oral squamous cell carcinoma [26]. These apparent discrepancies made us look with more detail into the tissue compartmentalization of VISTA+ and CD11b+ cells as well as CD8+ T-cells in MM bone marrow. These analyses revealed that localization of VISTA+ and CD11b+ cells in the tumor (but not stroma) was significantly correlated with the localization of CD8+ T-cells in the same compartment. This is likely to be due to the adaptivity of immune responses, in which an increase in CD8+ T-cell numbers is generally followed by an increase in the number of immune-suppressive cells [27,28]. Notably, densities of VISTA+ and CD11b+ cells in tumors in relation to those of CD8+ T-cells in the stroma of the same patients clearly separated MM patients with low versus high OS. In fact, the *VISTA-associated T-cell exclusion* score, in which we combined localization of VISTA+ and CD11b+ cells in tumor over stroma multiplied by localization of CD8+ T-cells in stroma over tumor, was significantly associated with short OS and outperformed the individual associations of either tumoral density of VISTA+, CD11b+ cells, or stromal density of CD8+ T-cells with short OS in MM. The observation that the presence of VISTA+ and CD11b+ cells in tumor over stroma was accompanied by the absence of CD8+ T-cells in tumor over stroma in case of low survival suggests that VISTA+ and CD11b+ cells may exclude entrance of newly arrived CD8 T-cells into tumors. Along this line, it would be interesting to assess whether the potentially adverse effects of CD11b+ myeloid-derived suppressor cell density toward recruitment of CD8+ T-cells depends on VISTA expression, its ligation with VSIG3 or PSGL1, and/or limited production of T-cell chemo-attractants.

While the number of patients for which gene expression was done was high and the HRs for VISTA were similar in the cumulative cohort as well as in the individual ones, the number of patients used for the multiplex IF staining in the exploratory cohort was small. Despite this shortcoming, the difference observed in the two patient groups when evaluating VISTA-associated T-cell exclusion was significant (Figure 6E, *p* < 0.0001). Taken together, we have demonstrated that VISTA gene expression in bone marrow aspirates is a strong, independent parameter for outcome in NDMM as well as RRMM, and that intra-tumoral CD11b myeloid cells represent a dominant cellular source for VISTA. Furthermore, we have defined the *VISTA-associated T-cell exclusion* score, which takes into account the localization of VISTA+ and CD11b+ cells in tumor and that of CD8+ T-cells in stroma, and showed that this score is associated with short survival in MM.

Our study is limited by a lack of validation and a relatively small patient sample size used for in situ stainings. Currently, we have reached the end of patient recruitment for a new randomized myeloma trial and once survival data are available, we will validate the *VISTA-associated T-cell exclusion* score in this large cohort, and at the same time, further delineate which cell type within CD11b+ cells shows the highest expression of VISTA. Furthermore, functional studies assessing the precise mechanism in which VISTA interferes with anti-tumor T-cell responses will help utilize this checkpoint in optimizing these responses and potentially use it for as a target for inhibition. Aside from these outlooks, we expect that introduction of the *VISTA-associated T-cell exclusion score* and enhanced understanding of CD8+ T-cell immunity in MM will facilitate future development of the blockade of VISTA, being in line with reports that point to VISTA as an actionable target in other cancers such as mesothelioma [1,8]. Testing of VISTA as a target in MM would address a high clinical need as IMiDs, monoclonal antibodies, and more recently, the use anti-BCMA CAR T-cell therapy, still go hand-in-hand with a high relapse rate.

## 4. Materials and Methods

### 4.1. Patients and Assessment of Clinical Responses

Biomaterials, gene expression data, and clinical data of patients with MM who entered one of eight trials, with a total number of patients 1654, were used to assemble a database (for details including treatment history, see Table 1). These trials with public accession and/or registration numbers included: HOVON-65/GMMG-HD4 (GSE19784; ISRCTN64455289); UAMS-TT2 (GSE2658; NCT00573391); MRC-IX non-intensive (GSE15695; ISRCTN68454111); APEX (GSE9782; registered under M34100–024, M34100–025 and NCT00049478/NCT00048230); HOVON-87/NMSG18 (EudraCT number 2007–004007-34); MRC-IX intensive (GSE15695); UAMS-TT3 (GSE15695; E-TABM-1138; NCT00081939); and UAMS-TT6 (GSE57317). For the purpose of multivariate analysis, the CoMMpass database of 718 patients was used (NCT01454297; research.mmrf.org, 24 March 2021; version IA13).

All trials had been previously approved by the responsible ethics committees and trial data are available according to the above-mentioned trial numbers.

Deidentified individual participant data are available indefinitely through the corresponding author.

### 4.2. Bone Marrow Aspirates and Gene Expression Analysis

Pretreatment bone marrow samples from MM patients were enriched for CD138 and used to determine whole genome expression. The different patient cohorts, the gene expression platforms (Affymetrix U133Plus2.0 A/B microarrays or RNA-Seq) as well as the usage of cohorts toward either discovery, validation, or second validation (*n* = 1045, 609, and 718 patients, respectively) are listed in Table 1. Gene expression data were processed using a Brain Array custom design file (CDF; Version 22.0.0, ENSG) that was mapped against the Ensemble Gene database [29]. Data were MAS5 normalized toward a default target value of 500, and resulting intensities were log2 transformed. Subsequently, batch normalization was applied by scaling expressions, so that all gene-based means and variances were equalized among patient cohorts. Genes not annotated and genes not expressed in individual cohorts (i.e., log2 expression < 8 in at least 50% of patients) were excluded from analysis. Analysis comprised a 3-staged and sequential approach that started with 366 immune-related genes that were each categorized into one of seven lists of genes covering different aspects of T-cell evasion such as antigen presentation, immune cells, cell death, immune checkpoints, metabolic checkpoints, oncogenic signaling pathways, and tumor micro-environment (see [15]). In the first step, we tested the seven gene lists for their significance toward OS according to ridge regression Cox models. In this step, the ridge shrinkage parameter was determined in a 10-fold cross validation by optimizing partial likelihood. The performance was determined in double loop cross testing with a leave-one-out outer loop in the discovery set, followed by determining the performance in the validation set. Second, for gene lists that survived the first step (i.e., at least one gene per list associated with OS for a list of genes to become significant), individual genes were tested for significance toward OS again using discovery and validation sets as well as a multivariate model in which stages according to the international staging system (ISS) were added. Third, individual genes that survived the second step (i.e., association with OS in multivariate model) were then analyzed in a multivariate model using a second validation set (i.e., the CoMMpass dataset), in which ISS stage and the time-dependent variable autologous stem cell transplantation were added (these parameters were available for all patients). Bonferroni-Holm multiple testing adjustments were applied in each of the above three steps [30].

### 4.3. Frequencies of Immune Cell Populations

Bone marrow aspirates used for gene expression analysis were assessed for the presence of non-tumor cells derived from the immune micro-environment. To this end, micro-array data of the different patient cohorts were used for deconvolution of 21 immune cell populations (LM22 excluding plasma cells) using the algorithm CIBERSORT [31]. In addition, the percentage of CD138+ cell purity in bone marrow samples following plasma cell-enrichment from the HOVON65/GMMG-HD4 and the HOVON87/NMSG18 datasets was analyzed as a confounding variable toward the outcomes of gene expression analysis.

### 4.4. Multiplexed Immunofluorescence (IF)

Multiplexed IF was performed on 4-μm trephine bone marrow sections from an exploratory patient cohort from the HOVON87/NMSG18 trial using antibodies against distinct immune cell subsets/markers in combination with OPAL reagents (Akoya Biosciences, Marlboro, MA, USA) (for specifics of antibodies and reagents, see Appendix A). Patients (*n* = 22) were divided into two groups with high- and low OS using the median value as a cut-off. Immune staining included multiple cycles of: antigen retrieval (15 min boiling in antigen retrieval buffer, pH 6 or pH 9 depending on primary antibodies) followed by cooling, blocking, and consecutive staining with primary antibodies, HRP-polymer, and Opal fluorophores; cycles were repeated until all markers were stained. Finally, nuclei were stained with DAPI.

### 4.5. Analysis of Multiplexed Immunofluorescence Images

Images were obtained of whole tissue sections using the Vectra^®^ 3.0 system (Akoya Biosciences, Menlo Park, CA, USA). Spectral unmixing of images was performed using the inForm^®^ software (Akoya Biosciences, Inc.) to visualize individual markers as well as autofluorescence, after which images were processed and analyzed using in-house written scripts (Python Software Foundation, v3.6.8). In short, images were subjected to background correction and normalization per channel. The foreground was defined as the normalized signal level exceeding 0.2 in any channel, and only foreground regions larger than 5000 px^2^ (293 μm^2^) were considered for further analysis. Next, images underwent tissue segmentation, to which end, data from the CD138 channel were passed through a Gaussian filter of 10px and signal levels exceeding 0.2 were assigned to the tumor region, whereas the rest were assigned to the stroma region, and both types of regions were subjected to the same size threshold as the foreground (293 μm^2^). Very large cells expressing VISTA protein, morphologically identical to megakaryocytes, were excluded from further analysis. Subsequently, nucleus identification was performed, where raw DAPI signals were background corrected, thresholded, and subjected to watershed segmentation to obtain the nuclei of single cells. These nuclei were then used to define cells through Voronoi segmentation of foreground signals using nuclei regions as seeds. Cell regions with areas between 500 to 5000 px^2^ (29.3 to 293 μm^2^) were taken along for phenotyping, to which end, averages of normalized intensity, and thresholds of 0.5 for CD138 and 1.5 for other channels (VISTA, CD8 and CD11b) were used to assign individual phenotypes. Cells were assigned to either tumor or stroma regions according to the location of the center of their nuclei, and distances between individual cells were determined according to the center of their areas. Cellular densities were calculated by dividing the number of cells with a certain phenotype by the total area of that region (i.e., tumor, stroma), and were averaged per patient across different image regions. Distances from a cell with a certain phenotype to the nearest cell with another phenotype were calculated by nearest neighbor analysis; this was done in the region of interest (i.e., tumor, stroma), and was averaged across all cells, and per patient across all image regions.

Cellular densities and distances between cells were assessed for their prognostic value by comparing MM patients with low versus high OS. In addition, preferential localizations of immune cells in a certain tissue compartment (i.e., ratios between tumoral and stromal densities of particular immune cells, or combinations thereof) were also assessed for their prognostic value.

### 4.6. Statistics

Ridge regression analysis was performed to determine significance toward OS of defined lists of genes, whereas Cox regression analysis was performed to determine significance toward OS of individual genes. Unpaired, two-tailed, Student’s *t*-test was used to compare densities of cells as well as distances between cells in bone marrow tissues, Spearman’s correlation was used to assess linear relationships between immune parameters, and the log-rank test was used to compare Kaplan–Meier curves. Differences were considered significant when *p* < 0.05, with the following levels of significance: * *p* < 0.05, ** *p* < 0.01, and *** *p* < 0.001.

## 5. Conclusions

Expression of VISTA in the tumor microenvironment is an independent prognostic factor for survival in both newly diagnosed as well as relapsed multiple myeloma. The main cellular source of VISTA in myeloma are CD11b+ myeloid cells. The combination of abundance of VISTA+ and CD11b+ cells in the tumor but not stromal tissue together with low presence of CD8+ T-cells in the same tissue compartment, termed a high *VISTA-associated T-cell exclusion* score, was significantly associated with short OS [HR: 16.6; 95% CI: 4.54–62.50; *p* < 0.0001].

## Figures and Tables

**Figure 1 cancers-13-02219-f001:**
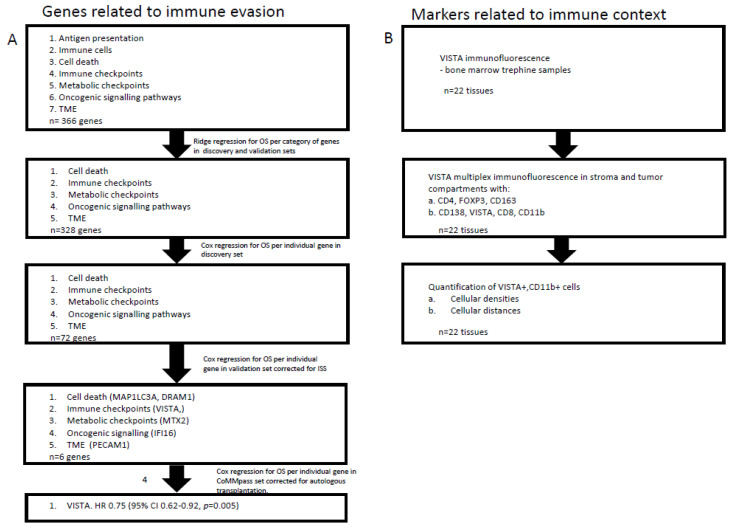
Chart illustrating workflow and showing analysis steps of gene expression and in situ staining. Individual steps of gene expression analysis and those of in situ stainings are depicted on the left and right-hand side, respectively.

**Figure 2 cancers-13-02219-f002:**
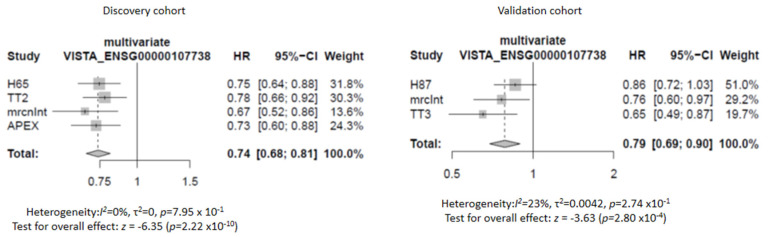
VISTA gene expression relates to OS in MM. Forrest plot showing protective Hazard Ratios (HR) of all trials except TT6 for which data on ISS stage was not available.

**Figure 3 cancers-13-02219-f003:**
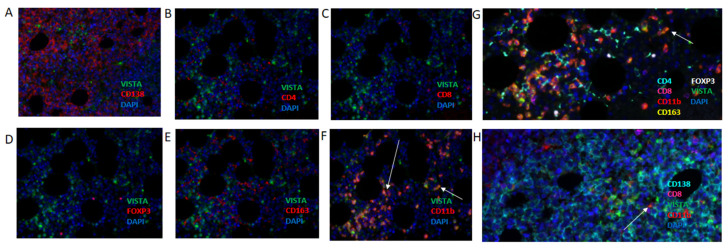
In MM, VISTA is expressed by CD11b+ cells, but not CD138+ cells, CD4+, CD8+, FOXP3 lymphocytes nor CD163+ macrophages. Multiplex immunofluorescence using bone marrow trephine slides with VISTA and CD138 (**A**); CD4 (**B**); CD8 (**C**); FOXP3 (**D**); CD163 (**E**); and CD11b (**F**). Co-expression of VISTA and each of these markers, only observed in case of CD11b, is indicated with arrows pointing at orange cells. (**G**) Multiplex showing combined stainings for CD138, CD11b, CD8, VISTA and DAPI. (**H**) Multiplex showing combined stainings for CD138, CD8, VISTA, CD11b and DAPI.

**Figure 4 cancers-13-02219-f004:**
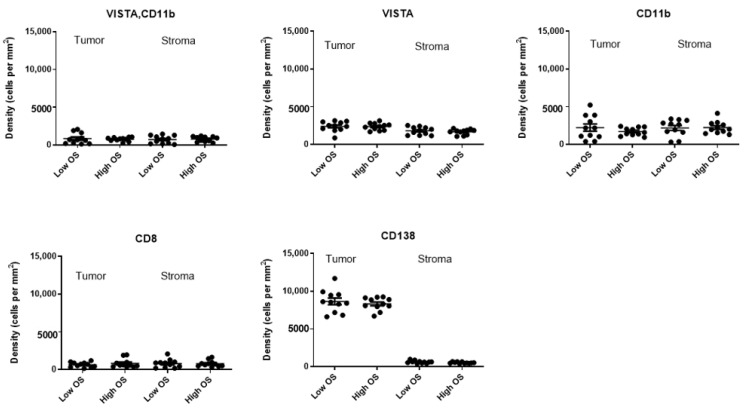
Densities of immune cells, including VISTA+, CD11b+ cells, do not differ between MM patients with low and high OS. Scatter plots of densities of cells (y-axis) in tumoral and stromal compartments of bone marrows from patients (*n* = 22) with low versus high OS. OS was separated according to median value; and lines indicate mean +/− SEM densities.

**Figure 5 cancers-13-02219-f005:**
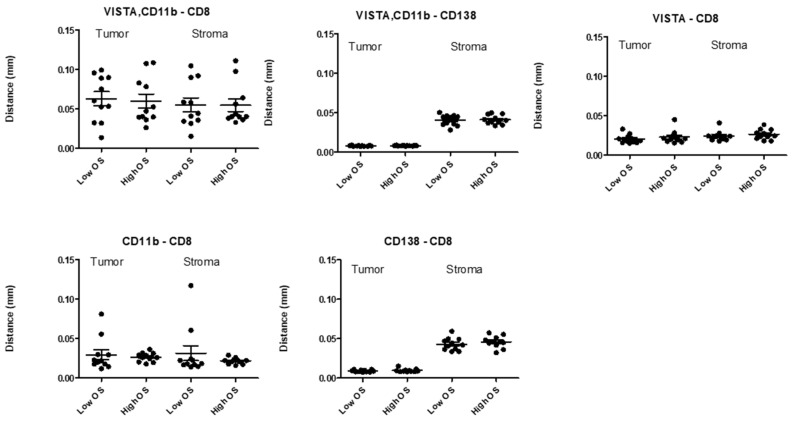
Distances between immune cells, including VISTA+, CD11b+ cells and CD8+ T-cells, do not differ between MM patients with low and high OS. Scatter plots of distances between cells (x-axis) and CD8 T-cells in tumoral and stromal compartments of bone marrows from patients (*n* = 22) with low versus high OS. OS was separated according to median value; and lines indicate mean +/− SEM distances.

**Figure 6 cancers-13-02219-f006:**
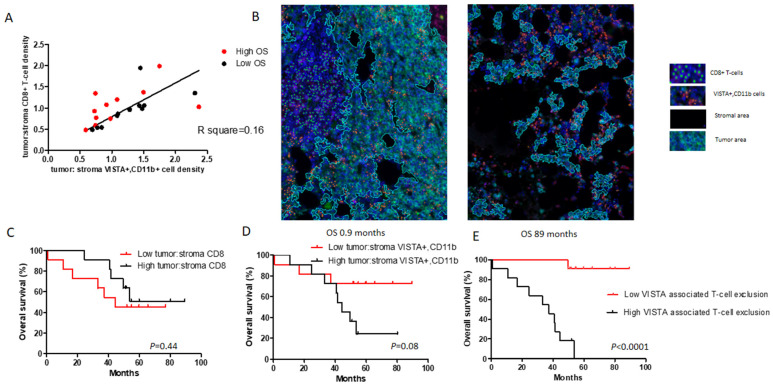
High abundance of VISTA+, CD11b+ cells combined with low abundance of CD8+ T-cells in tumor associates with low OS in MM. (**A**) Correlation plot between preferential CD8 T-cell density and preferential VISTA+, CD11b cell density (see Results for details). (**B**) Immunofluorescence image showing one patient with low OS (0.9 months, left-hand side) and another patient with high OS (89 months, right-hand side). Tumor area is encircled by cyan line. Relative abundance of VISTA+, CD11b+ cells (orange) in tumor area compared to CD8 T-cells (green) associates with low OS, whereas low abundance of VISTA+, CD11b cells in tumor area compared to CD8 T cells is associated with high OS. (**C**–**E**) Kaplan-Meier estimates showing OS for tumoral abundance of CD8 T-cells (**C**), VISTA+, CD11b+ cells (**D**), and the combination of these two parameters.

**Table 1 cancers-13-02219-t001:** Database covering 9 patient cohorts in MM listing number of patients, line of therapy, treatment arms and type of transcriptomic analysis performed.

Discovery *n* = 1045	*n*	Type	Treatment	Analysis Platform
HOVON65/GMMG-HD4	327	NDMM ^1^	PAD/VAD ^3^	Affymetrix Gene chip Plus 2.0
UAMS-TT2	345	NDMM	TD/VincristineDex ^4^	Affymetrix Gene chip Plus 2.0
MRC-IX non-IC	109	NDMM	CTDa/MP ^5^	Affymetrix Gene chip Plus 2.0
APEX	264	RRMM ^2^	BOR/DEX ^6^	Affymetrix Gene chip Plus A + B
Validation *n* = 609				
HOVON87/NMSG18	178	NDMM	MPT-T/MPR-R ^7^	Affymetrix Gene chip Plus 2.0
UAMS-TT3	238	NDMM	VTD ^8^	Affymetrix Gene chip Plus 2.0
MRC-IX intensive	138	NDMM	CTD/CVAD ^9^	Affymetrix Gene chip Plus 2.0
UAMS-TT6	55	RRMM	VTD	Affymetrix Gene chip Plus 2.0
2nd validation cohort *n* = 1000				
CoMMpass	718	NDMM/RRMM	Multiple 1st, 2nd and 3rd regimens	RNAseq

1. Newly diagnosed MM.; 2. Relapsed refractory MM; 3. Bortezomib/doxorubicin/dexamethsone and vincristine/doxorubicine/dexamethason.; 4. Thalidomide/dexamethason/vincristine; 5. Cyclophosphamide/thalidomide/dexamethasone/melphalan; 6. Bortezomib/dexamethasone; 7. Melphalan/prednisolone/thalidomide vs. melpahalan/prednisolone/lenalidomide; 8. Bortezomib/thalidomide/dexamethasone; 9. Cyclophosphamide/thalidomide/dexamethasone/melphalan/cyclophosfamide/vincristine/doxorubicin.

**Table 2 cancers-13-02219-t002:** Ridge regression analysis for OS per list of immune-evasive genes.

Categories of Immune Evasion	*p* Value Discovery			*p* Value Validation		
	Multivariate	Multivariate Holm	Passed	Multivariate	Multivariate Holm	Passed
Antigen presentation (*n* = 14)	0.92 × 10^−3^	9.20 × 10^−4^	Yes	1.50 × 10^−1^	1.50 × 10^−1^	No
Immune cells (*n* = 1)	2.78 × 10^−7^	5.57 × 10^−7^	Yes	6.00 × 10^−2^	1.20 × 10^−1^	No
Cell death (*n* = 51)	3.85 × 10^−20^	2.31 × 10^−19^	Yes	2.73 × 10^−7^	1.92 × 10^−6^	Yes
Immune checkpoints (*n* = 9)	1.27 × 10^−9^	3.81 × 10^−9^	Yes	2.76 × 10^−7^	1.92 × 10^−6^	Yes
Metabolic checkpoints (*n* = 77)	1.02 × 10^−20^	7.19 × 10^−20^	Yes	4.46 × 10^−6^	2.23 × 10^−5^	Yes
Oncogenic signaling pathways (*n* = 38)	4.18 × 10^−16^	2.09 × 10^−15^	Yes	1.71 × 10^−5^	6.23 × 10^−5^	Yes
TME (*n* = 20)	1.22 × 10^−15^	4.91 × 10^−15^	Yes	1.53 × 10^−5^	6.22 × 10^−5^	Yes

**Table 3 cancers-13-02219-t003:** Cox regression analysis for OS per individual genes.

Categories of Immune Evasion	Gene ^1^	HR	95% CI	*p* Value
Cell death	MAP1LC3A	0.80	0.71–0.90	0.001
Cell death	DRAM1	0.79	0.70–0.90	0.017
Immune checkpoints	VISTA	0.76	0.67–0.86	0.001
Metabolic checkpoints	MTX2	1.30	1.14–1.48	0.003
Oncogenic signaling pathways	IFI16	1.37	1.21–1.55	0.0005
TME	PECAM1	0.73	0.64–0.82	0.0001

^1^ Genes listed are those that are significantly associated with OS after multivariate analysis.

**Table 4 cancers-13-02219-t004:** Cox regression analysis for OS per individual gene in CoMMpass database.

Categories of Immune Evasion	Gene	HR	95% CI	*p* Value
Cell death	MAP1LC3A	0.87	0.69–1.1	0.25
Cell death	DRAM1	0.86	0.69–1.1	0.19
Cell death	BCL2	0.87	0.7–1.1	0.19
Immune checkpoints	CD40	0.81	0.64–1.0	0.07
Immune checkpoints	VISTA	0.75	0.62–0.92	0.005
Metabolic checkpoints	MTX2	0.88	1.7–3.4	0.22
Oncogenic signaling pathways	IFI16	1.1	0.88–1.3	0.48
TME	N/A	N/A	N/A	N/A

PECAM1 data was not available in the CoMMpass dataset.

## Data Availability

No new data were created or analyzed in this study. Data sharing is not applicable to this article.

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
