# Peer review of "V-Domain Ig Suppressor of T Cell Activation (VISTA) Expression Is an Independent Prognostic Factor in Multiple Myeloma"

_cancers, 2021, doi:10.3390/cancers13092219_

Round 1

Reviewer 1 Report

The Authors report a study on the mechanisms of escape from T-cell immunity during the progression of multiple myeloma disease by analyzing the expression of genes related to T-cell evasion. The paper is well written and the reported results have relevance in the comprehension of the mechanisms related to MM progression and in MM therapy. I believe that the manuscript can be accepted for publication in this Journal after minor revisions.

In particular, I believe that the Authors should discuss in the Introduction section the importance of VISTA inhibition as a new approach to cancer therapy. I would suggest the Authors to consider some recent literature data such as: S. Muller et al., Mod Pathol., 2020, 33, 303-311, doi: 10.1038/s41379-019-0364-z; A. Boujmia, Naunyn-Schmiedeberg's Arch. Pharmacol., 2021, doi: 10.1007/s00210-021-02068-4.

Moreover, in the Conclusions section, the relevance of the reported results in MM treatment should be better discussed.

Author Response

Response:

Authors thank R1 for reviewing the manuscript and communicating constructive feedback. The discovery of VISTA as a prognostic factor in MM was not hypothesized beforehand but rather an outcome of our study into determining the factors that regulate anti-tumor T-cell immunity in MM. The publications you bring forward regarding the blockade of VISTA are welcomed, and used towards a more elaborate interpretation regarding the clinical relevance of our findings (i.e., blockade of VISTA in MM) in the discussion section. In addition, we have added separate, more general text regarding the importance of checkpoint molecules in the introduction section.

The above changes are now mentioned (highlighted in yellow) in the Introduction (line #28 to 30) and Discussion sections (line #252 to 258) of the revised manuscript.

Reviewer 2 Report

cancers-1176054

V-domain Ig suppressor of T cell activation (VISTA) expression is an independent prognostic factor in multiple myeloma.

The article " V-domain Ig suppressor of T cell activation (VISTA) expression is an independent prognostic factor in multiple myeloma" (cancers-1176054) by Mutsaers P, et al. demonstrated that VISTA predicted clinical outcome in analysis using transcriptomic data of bone marrow from 8 clinical trials. In addition, the author showed that VISTA+, CD11b+ and CD8+ cells in tumor compartment predicted OS in the myeloma patients. These data were supposed to be very informative and suggestive for using PI. However, there are several major and minor issues to be addressed as below.

Major issues

  1. The author analyzed the clinical significance of VISTA, CD11b+, and CD8+ cells using the bone marrow samples from 22 of myeloma patients. I considered that the results were affected by the kind of anti-myeloma agents, especially IMiDs. The author should describe the patient characteristics of 22 myeloma patients in detail.

Minor issues

  1. The author showed that VISTA was positive on CD11b positive cells in figure 3. In general, CD11b expresses on macrophage, NK cells, and granulocyte. The author should add the data about staining of CD68, CD56, and CD13 in order to identify with what kind of blood cells were both VISTA and CD11b positive if possible.

  1. In figure 5, there was not a significant relation between OS and the distances from VISTA, CD11b+ to CD8+ cells, and from CD8+ to CD138+ cells. However, the figure 5 seemed to show that there was a difference between location of cells (tumor vs stroma) and the distance of these cells. If there was statistical significance, you could discuss about that.

  1. The author mentioned that VISTA blockade enhanced T cell activity among the PD-L1 high expressed tumor cells independently from VISTA expression level in discussion. Was the expression of PD-L1 associated with clinical outcome and expression of VISTA in your cohort?

  1. Line 218. RDMM might be misspelled. The author should revise into “NDMM”.

Author Response

Response to major comment #1:

Authors thank R2 for reviewing the manuscript and pointing out omission of treatment data of the 22 patients from whom tumors were used for in situ stainings. Along R2’s recommendation, we have revised our manuscript, and included a supplementary table with the requested patient characteristics (also included in this rebuttal). In this table, we have listed age, sex, treatment, VISTA mRNA expression, VISTA-associated T-cell exclusion score and other acknowledged prognostic variables, such as cytogenetics, serum albumin and b2 microglobulin. Of interest, when the 2 treatment arms (see table below) were tested for VISTA mRNA expression (2-tailed Student’s t-test; p=0.07; 95%CI 0.06-1.35) or VISTA-associated T-cell exclusion score (p=0.29; 95%CI 0.16-0.51), no significant differences were observed. With respect to IMiD’s, these drugs were present in both arms. It is noteworthy, however, that lenalidomide in the MPR arm is a much more potent drug than thalidomide in the MPT arm, which strongly suggests that in case IMiD would be correlated with VISTA expression or function, a difference between both treatment arms would have been detected. Moreover, the APEX trial (included in the gene expression analysis) did not enroll IMiD-treated patients, yet in this trial outcomes regarding VISTA expression were similar to that of other trials (Figure 2 of manuscript), again suggesting that treatment did not confound our results.

The clinical characteristics of the 22 patients used for in situ staining are detailed in the new Supplementary Table 4 and referred to in Results section (line #80 to 84) of the revised manuscript.

Response to minor comment 1:

R2 makes a relevant suggestion regarding the delineation of the exact cellular source of VISTA-expressing CD11b+ cells. Additional in situ stainings as proposed by R2 are currently challenged by limited availability of tissue samples. We have, however, as now indicated in the last sentence of our revised manuscript(Line #246 to 249), reached the end of patient recruitment for another randomized trial and, in addition to validating our findings, will be studying these subsets in this cohort.

Response to minor comment 2:

As expected, density of CD138 was higher and distances to CD138 were lower in tumor region irrespective of survival. There was no statistical significant association between these parameters. There is a correlation between the distance of cells from CD138 cells in different compartments but merely turned out due to the ‘crowding’ of CD138 cells in tumor, therefor, decreasing intercellular distances

Response to minor comment 3:

PDL1 was one of the genes that was tested as part of the immune checkpoint category of genes. Although it was significantly associated with OS in the discovery set, it did NOT reach statistical significance in the validation cohort after correction for multiple testing HR=0.80 p(holm)=0.062.. Along R2’s remark, we have now mentioned this trend, and discussed this finding in the Discussion section of the revised manuscript (line #182 to 184).

Response to minor comment 4:

We thank the reviewer for bringing this to our attention. This was indeed a spelling mistake and is now corrected to NDMM in the revised manuscript.

Round 2

Reviewer 2 Report

I considered that any revisions was not needed and this article was suitable for "Cancers".